# CONFIDENT SINKHORN ALLOCATION FOR PSEUDO-LABELING

## ABSTRACT

Semi-supervised learning is a critical tool in reducing machine learning's dependence on labeled data. It has been successfully applied to structure data, such as image and language data, by exploiting the inherent spatial and semantic structure therein with pretrained models or data augmentation. Some of these methods are no longer applicable for the data where domain structures are not available because the pretrained models or data augmentation can not be used.

Due to simplicity, existing pseudo-labeling (PL) methods can be widely used without any domain assumption, but are vulnerable to noise samples and to greedy assignments given a predefined threshold which is typically unknown. This paper addresses this problem by proposing a Confident Sinkhorn Allocation (CSA), which assigns labels to only samples with high confidence scores and learns the best label allocation via optimal transport. CSA outperforms the current state-of-the-art in this practically important area of semi-supervised learning.

## 1 INTRODUCTION

The impact of machine learning continues to grow in fields as disparate as biology (Libbrecht & Noble, 2015; Tunyasuvunakool et al., 2021), quantum technology (Biamonte et al., 2017; van Esbroeck et al., 2020; Nguyen et al., 2021), brain stimulation (Boutet et al., 2021; van Bueren et al., 2021), and computer vision (Esteva et al., 2021; Yoon et al., 2022). Much of this impact depends on the availability of large numbers of annotated examples for the machine learning models to be trained on. The data annotation task by which such labeled data is created is often expensive, and sometimes impossible, however. Rare genetic diseases, stock market events, and cyber-security threats, for example, are hard to annotate due to the volumes of data involved, the rate at which the significant characteristics change, or both.

**Related work.** Fortunately, for some classification tasks, we can overcome a scarcity of labeled data using semi-supervised learning (SSL) (Zhu, 2005; Huang et al., 2021; Killamsetty et al., 2021; Olsson et al., 2021). SSL exploits an additional set of unlabeled data with the goal of improving on the performance that might be achieved using labeled data alone (Lee et al., 2019; Carmon et al., 2019; Ren et al., 2020; Islam et al., 2021).

Domain specific: Semi-supervised learning for image and language data has made rapid progress (Oymak & Gulcu, 2021; Zhou, 2021; Sohn et al., 2020) largely by exploiting the inherent spatial and semantic structure of images (Komodakis & Gidaris, 2018) and language (Kenton & Toutanova, 2019). This is achieved typically either using pretext tasks (*eg.* (Komodakis & Gidaris, 2018; Alexey et al., 2016)) or contrastive learning (*eg.* (Van den Oord et al., 2018; Chen et al., 2020)). Both approaches assume that specific transformations applied to each data element will not affect the associated label.

Greedy pseudo-labeling: Without domain assumption, a simple but effective way for SSL is pseudo-labeling (PL) (Lee et al., 2013) which generates 'pseudo-labels' for unlabeled samples using a model trained on labeled data. A label $k$ is assigned to an unlabeled sample $\mathbf{x}_i$ where a predicted class probability is larger than a predefined threshold $\gamma$ as

$$y_i^k = \mathbb{1}\left[p(y_i = k \mid \mathbf{x}_i) \geq \gamma\right] \tag{1}$$

Table 1: Comparison with the related approaches in terms of properties and their relative trade-offs.

| Algorithms | Not domain specific | Uncertainty consideration | Non-greedy |
|---|---|---|---|
| Pseudo-Labeling (Lee et al., 2013) | ✓ | ✗ | ✗ |
| FlexMatch (Zhang et al., 2021) | ✓ | ✗ | ✗ |
| Vime (Yoon et al., 2020) | ✗ | ✗ | NA |
| MixMatch (Berthelot et al., 2019) | ✗ | ✗ | NA |
| FixMatch (Sohn et al., 2020) | ✗ | ✗ | NA |
| UPS (Rizve et al., 2021) | ✓ | ✓ | ✗ |
| SLA (Tai et al., 2021) | ✓ | ✗ | ✓ |
| **CSA** | ✓ | ✓ | ✓ |

where $\gamma \in [0,1]$ is a threshold used to produce hard labels and $p(y_i = k \mid \mathbf{x}_i)$ is the predictive probability of the $i$-th data point belonging to the class $k$-th. A classifier can then be trained using both the original labeled data and the newly pseudo-labeled data. Pseudo labeling is naturally an iterative process, with the next round of pseudo-labels being generated using the most-recently trained classifier. The key advantage of pseudo-labeling is that it does not inherently require any domain assumption and can be generally applied to most domains, including tabular data.

Greedy PL with uncertainty: Rizve et al. (2021) propose an uncertainty-aware pseudo-label selection (UPS) that aims to reduce the noise in the training process by using the uncertainty score – together with the probability score for making assignments:

$$y_i^k = \mathbb{1}\Big[ p(y_i = k \mid \mathbf{x}_i) \geq \gamma \Big] \, \mathbb{1}\Big[ \mathcal{U}\big( p(y_i = k \mid \mathbf{x}_i) \big) \leq \gamma_u \Big] \tag{2}$$

where $\gamma_u$ is an additional threshold on the uncertainty level and $\mathcal{U}(p)$ is the uncertainty of a prediction $p$. As shown in Rizve et al. (2021), selecting predictions with low uncertainties greatly reduces the effect of poor calibration, thus improving robustness and generalization.

However, the aforementioned works in PL are *greedy* in assigning the labels by simply comparing the prediction value against a predefined threshold $\gamma$ irrespective of the relative prediction values across samples and classes. Such greedy strategies will be sensitive to the choice of a threshold.

Non-greedy pseudo-labeling: FlexMatch (Zhang et al., 2021) considers adaptively selecting a threshold $\gamma_k$ for each class based on the level of difficulty. This threshold is adapted using the predictions across classes. However, the selection process is still heuristic in comparing the prediction score with an adjusted threshold. Recently, Tai et al. (2021) provide a novel view in connecting the pseudo-labeling assignment task to optimal transport problem, called SLA, which inspires our work. SLA and FlexMatch are better than existing PL in that their *non-greedy* label assignments not only use the single prediction value but also consider the relative importance of this value across rows and columns in a holistic way. However, both SLA and FlexMatch can overconfidently assign labels to noise samples and have not considered utilizing uncertainty values in making assignments.

**Contributions.** We propose here a semi-supervised learning method that does not require any domain-specific assumption for the data. We hypothesize that this is by far the most common case for the vast volumes of data that exist. The method we propose is based on pseudo-labeling of a set of unlabeled data using Confident Sinkhorn Allocation (CSA). Our method is theoretically driven by the role of uncertainty in robust label assignment in SSL. CSA utilizes Sinkhorn's algorithm (Cuturi, 2013) to assign labels to only the data samples with high confidence scores. By learning the label assignment with optimal transport, CSA eliminates the need to predefine the heuristic thresholds used in existing pseudo-labeling methods, which can be greedy. The proposed CSA is applicable to any data domain, and could be used in concert with consistency-based approaches (Sohn et al., 2020), but is particularly useful for data domain where pretext tasks and data augmentation are not applicable, such a tabular data.

## 2 CONFIDENT SINKHORN ALLOCATION (CSA)

We consider the semi-supervised learning setting whereby we have access to a dataset consisting of labeled examples $\mathcal{D}_l = \{\mathbf{x}_i, y_i\}_{i=1}^{N_l}$, and one of unlabeled examples $\mathcal{D}_u = \{\mathbf{x}_i\}_{i=1}^{N_u}$ where $\mathbf{x}_i \in \mathbb{R}^d$ and $y_i \in \mathcal{Y} = \{1, \ldots, K\}$. We define also $\mathcal{X} = \{\mathbf{x}_i\}, i = \{1, \ldots, N_l + N_u\}$. Our goal is to utilize $\mathcal{D}_l \cup \mathcal{D}_u$ to

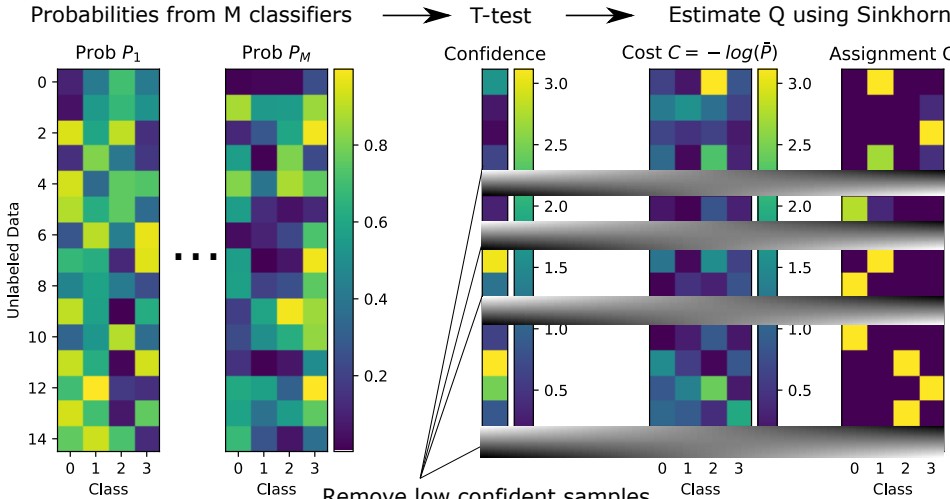

Figure 1: A depiction of CSA in application. We estimate the ensemble of predictions $P$ on unlabeled data using $M$ classifiers which can result in different probabilities. We then identify high-confidence samples by performing a T-test. Next, we estimate the label assignment $Q$ using Sinkhorn's algorithm. The cost $C$ of the optimal transport problem is the negative of the probability averaging across classifiers, $C = -\log \bar{P}$. We repeat the process on the remaining unlabeled data as required.

learn a predictor $f : \mathcal{X} \to \mathcal{Y}$ that is more accurate than a predictor trained using labeled data $\mathcal{D}_l$ alone. A notation summary is provided in Appendix Table 3.

Generating high-quality pseudo labels is critical to the final classification performance, as erroneous label assignment can quickly lead the iterative pseudo-labeling process astray. We provide in Sec. 2.1 a theoretical analysis of the role and impact of uncertainty in pseudo-labeling, the first such analysis as far as we are aware. Based on the theoretical result, we propose two approaches to identify the high-confidence samples for assigning labels and use the Sinkhorn's algorithm to find the best label assignment. We name the proposed algorithm Confident Sinkhorn Allocation (CSA). We provide a diagram demonstrating CSA in Fig 1, and a comparison against related algorithms in Table 1.

## 2.1 THEORETICAL ANALYSIS ON THE EFFECT OF UNCERTAINTY IN PSEUDO-LABELING

Our theoretical results highlight two properties of PL settings: (i) less noise and uncertainty are beneficial, and (ii) more unlabeled data is beneficial. For the analysis, we make a popular assumption that input features $\mathbf{x}_i \in \mathcal{D}_l$ and $\mathcal{D}_u$ are sampled i.i.d. from a feature distribution $P_X$, and the labeled data pairs $(\mathbf{x}_i, y_i)$ in $\mathcal{D}_l$ are drawn from a joint distribution $P_{X,Y}$.

We consider the binary classification problem in one-dimensional space as follows: generate the feature $x \mid y = +1 \overset{\text{iid}}{\sim} \mathcal{N}(\mu_+, \sigma^2)$ and similarly $x \mid y = -1 \overset{\text{iid}}{\sim} \mathcal{N}(\mu_-, \sigma^2)$, where $\mu_+$ and $\mu_-$ are the means of the positive and negative classes respectively, $\sigma$ is the standard deviation indicating the level of *data noisiness*. Without loss of generality, let $\mu_+ > \mu_-$. The optimal Bayes's classifier is $f(x) = \text{sign}\left(x - \frac{\mu_+ + \mu_-}{2}\right)$, *eg.* classify $y_i = +1$ if $x_i > (\mu_+ + \mu_-)/2$. Therefore, in the following, we measure the probability bound to learn $(\mu_+ + \mu_-)/2$ as a criterion for achieving good performance.

Let $\{\tilde{X}_i^+\}_{i=1}^{\tilde{n}_+}$ and $\{\tilde{X}_i^-\}_{i=1}^{\tilde{n}_-}$ be the sets of unlabeled data whose pseudo-labels are $+1$ and $-1$, respectively. Let $\{I_i^+\}_{i=1}^{\tilde{n}_+}$ be the binary indicators of correct assignments, such as if $I_i^+ = 1$, then $\tilde{X}_i^+ \sim \mathcal{N}(\mu_+, \sigma^2)$ and otherwise $\tilde{X}_i^- \sim \mathcal{N}(\mu_-, \sigma^2)$. Similarly, we define $\{I_i^-\}_{i=1}^{\tilde{n}_-}$ indicating correct assignment for negative class. Instead of generating from Bernoulli distribution as used in Yang & Xu (2020), we generate the binary indicators $I_i^+$ and $I_i^-$ from a classifier and denote the expectations $\mathbb{E}(I_i^+), \mathbb{E}(I_i^-)$ and variances $\text{Var}(I_i^+), \text{Var}(I_i^-)$, respectively. In the uncertainty estimation literature (Der Kiureghian & Ditlevsen, 2009), $\sigma^2$ is called the *aleatoric uncertainty* (noise from the observations) and $\text{Var}(I_i^+), \text{Var}(I_i^-)$ are the *epistemic uncertainties* (from the model knowledge). As mentioned above, we aim to learn $\frac{\mu_+ + \mu_-}{2}$, via the extra unlabeled data. It is natural from the binary

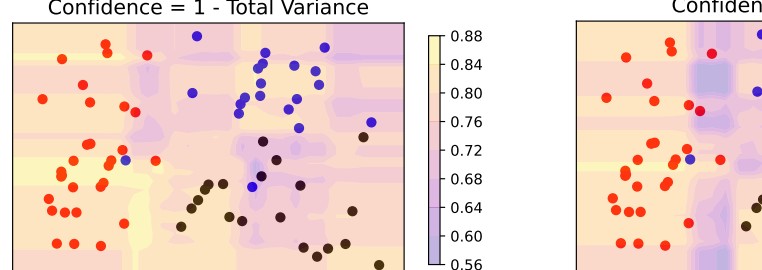

Figure 2: Examples of using total variance and T-test for estimating the confidence level on $K = 3$ classes. The yellow area indicates the highest confidence. We exclude the samples falling into the dark area where the value of T-test less than 2. We have simplified the tabular data (with possibly mixed categorical-continous-text features) into the $d = 2$ dimensional representation.

classification setting above to construct our estimate as $\hat{\theta} = \frac{1}{2}\left(\sum_{i=1}^{\tilde{n}_+} \tilde{X}_i^+/\tilde{n}_+ + \sum_{i=1}^{\tilde{n}_-} \tilde{X}_i^-/\tilde{n}_-\right)$. Please see Appendix A.1 for details.

**Theorem 1.** *For $\delta > 0$, with a probability at least $1 - 2\exp\left\{-\frac{2\delta^2[\tilde{n}_+ + \tilde{n}_-]}{9\sigma^2[\tilde{n}_+ \tilde{n}_-]}\right\} - \frac{9\left[Var(I_i^+) + Var(I_i^-)\right]}{4\delta^2}$, our estimate $\hat{\theta}$ satisfies $\left|\hat{\theta} - \frac{\Delta(\mu_+ - \mu_-)}{2} - \frac{(\mu_+ + \mu_-)}{2}\right| \leq \delta$ where $\Delta = |\mathbb{E}(I_i^+) - \mathbb{E}(I_i^-)|$.*

*Proof.* Our proof (see Appendix A.1) extends the result from Yang & Xu (2020) taking into account the data noisiness $\sigma^2$ and uncertainty $Var(I_i^+), Var(I_i^-)$ for pseudo-labeling. □

**Interpretation.** The theoretical result reveals several interesting aspects. First, training data imbalance affects the accuracy of our estimation, as discussed in Yang & Xu (2020). The more imbalanced the data is, the larger the gap $\Delta$ would be, which influences the closeness between the estimated $\hat{\theta}$ and desired value $\frac{(\mu_+ + \mu_-)}{2}$. Note that data imbalance is not the focus of this paper. In addition, the second term of the bound $\frac{\tilde{n}_+ + \tilde{n}_-}{\tilde{n}_+ \tilde{n}_-}$ shows that more unlabeled data $\tilde{n}_+ \uparrow$ or $\tilde{n}_- \uparrow$ is helpful for a good estimation. We empirically validate this property in Fig. 5. Finally, our analysis takes a step further to show that both aleatoric uncertainty $\sigma^2$ and epistemic uncertainty $Var(I_i^+) + Var(I_i^-)$ can reduce the probability of obtaining a good estimation. In other words, less uncertainty is more helpful. Note that the interpretation through the uncertainty has not been explored in Yang & Xu (2020). To the best of our knowledge, we are *the first* to theoretically characterize the role of data noise $\sigma^2$ and uncertainties $Var(I_i^+) + Var(I_i^-)$ affecting the performance in SSL.

## 2.2 IDENTIFYING HIGH-CONFIDENT SAMPLES FOR ASSIGNING LABELS

In Theorem 1, we show that the data sample noise and uncertainty can worsen the estimated classifier. Therefore, we propose to identify and ignore the high-uncertain samples from being assigned labels.

It is notorious that machine learning predictions can vary with different choices of hyperparameters. Under these variations, it is unclear to identify the most probable class to assign labels for some data points. Not only the predictive values but also the ranking order of these predictive samples vary with different hyperparameters. The variations in the prediction can be explained due to (i) the uncertainties of the prediction coming from the noise observations (aleatoric uncertainty) and model knowledge (epistemic uncertainty) (Hüllermeier & Waegeman, 2019) and (ii) the gap between the highest and second-highest scores is small. These bring a challenging fact that the highest score class can be changed with a different set of hyperparameters. This leads to the confusion for assigning pseudo labels because (i) the best set of hyperparameter is unknown given limited labeled data and (ii) we consider ensemble learning setting wherein multiple models are used together.

To address the two challenges above, we are motivated by the theoretical result in Theorem 1 to only assign labels to samples in which the most probable class is statistically significant than the second-most probable class. For this purpose, we propose to use two criteria for measuring the confidence level: (i) Welch's T-test and (ii) total variance across classes.

---

**Algorithm 1** Confident Sinkhorn Label Allocation

---

**Input:** lab data $\{\mathbf{X}_l, y_l\}$, unlab data $\mathbf{X}_u$, Sinkhorn reg $\varepsilon > 0$, fraction of assigned label $\rho \in [0,1]$
**Output:** Allocation matrix $Q \in \mathbb{R}^{N \times K}$

1 **for** $t = 1, ..., T$ **do**
2     Train $M$ models $\theta_1, ..., \theta_M$ given the limited labelled data $\mathbf{X}_l, y_l$
3     Obtain a confidence set $\mathbf{X}'_U \subset \mathbf{X}_U$ using T-value from Eq. (3) where $N := |\mathbf{X}'_U|$
4     Define a cost matrix $C$ from Eq. (16) using $\mathbf{X}'_U$
5     Set marginal distributions $\mathbf{r} = [\mathbf{1}_N^T; N(b_+^T \mathbf{1}_K - \rho b_-^T \mathbf{1}_K)]$ and $\mathbf{c} = [b_+N; N(1 - \rho b_-^T \mathbf{1}_K)]$
     `/* Sinkhorn's algorithm`                                                `*/`
6     **while** $j = 1, ..., J$ *or until converged* **do**
7        Initialize $a^{(j)} = \mathbf{1}_K^T$;    Update $b^{(j+1)} = \frac{\mathbf{r}}{\exp(-C_{i,k}/\varepsilon)a^{(j)}}$ and $a^{(j+1)} = \frac{\mathbf{c}}{\exp(-C_{i,k}/\varepsilon)^T b^{(j+1)}}$
8     Obtain $Q = \mathrm{diag}(a^{(J)}) \exp(-C_{i,k}/\varepsilon) \mathrm{diag}(b^{(J)})$          `// the assignment matrix`
9     Augment the assigned labels to $\{\mathbf{X}_l, y_l\}$ and remove these points from the unlabeled data $\mathbf{X}_u$.

---

**Welch's T-test.** For each data point $i$, we define two empirical distributions (see Appendix Fig. 6) of predicting the highest $\mathcal{N}(\mu_{i,\diamond}, \sigma_{i,\diamond}^2)$ and second-highest class $\mathcal{N}(\mu_{i,\oslash}, \sigma_{i,\oslash}^2)$,[1] estimated from the predictive probability across $M$ classifiers, such as $\mu_{i,\diamond} = \frac{1}{M}\sum_{m=1}^{M} p_m(y = \diamond \mid \mathbf{x}_i)$ and $\mu_{i,\oslash} = \frac{1}{M}\sum_{m=1}^{M} p_m(y = \oslash \mid \mathbf{x}_i)$ are the empirical means; $\sigma_{i,\diamond}^2$ and $\sigma_{i,\oslash}^2$ are the variances. We consider Welch's T-test (Welch, 1947) to compare the statistical significance of two empirical distributions:

$$\text{T-value}(\mathbf{x}_i) = \frac{\mu_{i,\diamond} - \mu_{i,\oslash}}{\sqrt{(\sigma_{i,\diamond}^2 + \sigma_{i,\oslash}^2)/M}}. \tag{3}$$

The degree of freedom for the above statistical significance is $\frac{(M-1)(\sigma_1^2+\sigma_2^2)^2}{\sigma_1^4+\sigma_2^4}$. As a standard practice in statistical testing (Neyman & Pearson, 1933; Fisher, 1955), we calculate the degree of freedom and look at the T-value distribution. In this particular setting, we get the following rules if the T-value is less than 2, the two considered classes are from the same distribution – *eg.*, the sample might fall into the dark area in *right* Fig. 2. Thus, we exclude a sample from assigning labels when its T-value is less than 2.

The estimation of the T-test above encourages separation between the highest and second-highest classes relates to entropy minimization (Grandvalet & Bengio, 2004) which encourages a classifier to output low entropy predictions on unlabeled data.

**Total variance.** In multilabel classification settings (Kapoor et al., 2012), multiple high score classes can be considered together. Thus, the Welch's T-test between the highest and second-highest is no longer applicable. We consider the following total variance across classes as the second criteria. This total variance for uncertainty estimation has also been used in UPS (Rizve et al., 2021) using Monte Carlo dropout (Gal & Ghahramani, 2016) in the context of deep neural network:

$$\mathcal{V}[p(y \mid \mathbf{x})] \approx \frac{1}{K}\sum_{k=1}^{K} \Big[ \underbrace{\frac{1}{M}\sum_{m=1}^{M}\Big(p_m(y = k \mid \mathbf{x}) - \sum_{m=1}^{M}\frac{p_m(y = k \mid \mathbf{x})}{M}\Big)^2}_{\text{variance of assigning } \mathbf{x} \text{ to class } k} \Big]. \tag{4}$$

We exclude the data point with high uncertainty measured by the total variance. It makes sense because a consensus of multiple classifiers is generally a good indicator of the labeling quality. In our setting, a high consensus is represented by low variance or high confidence. While the T-test naturally has a threshold of 2 to reject a data point, the total variance does not have such a threshold and thus we need to impose our own value. In the experiment, we reject 50% of the points with a high total variance score. We visualize the confidence estimated by T-test and total variance in Fig. 2.

We further consider the entropy criteria proposed in Malinin et al. (2020) as another way of estimating the confidence score in Appendix B.1.

**Discussion.** In Appendix Table 4, we empirically compare the performance using different uncertainty choices. We show that the Welch T-test is the best for this pseudo-labeling problem which can

---

[1] Denote the highest score class by $\diamond := \diamond(i)$ and second-highest score class by $\oslash := \oslash(i)$ for a data point $i$.

explicitly capture the uncertainty gap in predicting the highest and second-highest classes while total variance and entropy can not.

## 2.3 OPTIMAL TRANSPORT ASSIGNMENT

We use the confidence scores to filter out the uncertain data points and assign labels to the remaining using Sinkhorn algorithm (Cuturi, 2013). Particularly, we follow the novel view in Tai et al. (2021) to interpret the label assignment process as an optimal transportation problem between examples and classes, wherein the cost of assigning an example to a class is dictated by the predictions of the classifier.

Let us denote $N \leq N_u$ be the number of accepted points, *eg.* by T-test, from the unlabeled set. Let define an assignment matrix $Q \in \mathbb{R}^{N \times K}$ of $N$ unlabeled data points to $K$ classes such that we assign $\mathbf{x}_i$ to a class $k$ if $Q_{i,k} > 0$. We seek an assignment $Q$ that minimizes the total assignment cost $\sum_{ik} Q_{ik} C_{ik}$ where $C_{ik}$ is the cost of assigning an example $i$ to a class $k$ given by the corresponding negative probability as used in Tai et al. (2021), i.e., $C_{ik} := -\log p(y_i = k \mid \mathbf{x}_i)$ where $0 \leq p(y_i = k \mid \mathbf{x}_i) \leq 1$

$$\text{minimize}_Q \quad \langle Q, C \rangle \quad (5) \qquad \text{minimize}_{Q, \mathbf{u}, \mathbf{v}, \tau} \quad \langle Q, C \rangle \quad (10)$$

$$\text{s.t.} \quad Q_{ik} \geq 0 \quad (6) \qquad \qquad \text{s.t.} \quad Q_{ik} \geq 0, \mathbf{u} \succeq 0, \mathbf{v} \succeq 0, \tau \geq 0 \quad (11)$$

$$Q\mathbf{1}_K \leq \mathbf{1}_N \quad (7) \qquad \qquad Q\mathbf{1}_K + \mathbf{u} = \mathbf{1}_N \quad (12)$$

$$Q^T \mathbf{1}_N \leq N\mathbf{w}_+ \quad (8) \qquad \qquad Q^T \mathbf{1}_N + \mathbf{v} = \mathbf{w}_+ N \quad (13)$$

$$\mathbf{1}_N^T Q \mathbf{1}_K \geq N\rho \mathbf{w}_-^T \mathbf{1}_K \quad (9) \qquad \qquad \mathbf{u}^T \mathbf{1}_N + \tau = N(1 - \rho \mathbf{w}_-^T \mathbf{1}_K) \quad (14)$$

$$\mathbf{v}^T \mathbf{1}_K + \tau = N(\mathbf{w}_+^T \mathbf{1}_K - \rho \mathbf{w}_-^T \mathbf{1}_K) \quad (15)$$

where $\mathbf{1}_K$ and $\mathbf{1}_N$ are the vectors one in $K$ and $N$ dimensions, respectively; $\rho \in [0, 1]$ is the fraction of assigned label, i.e., $\rho = 1$ is full allocation; $\mathbf{w}_+, \mathbf{w}_- \in \mathbb{R}^k$ are the vectors of upper and lower bound assignment per class which can be estimated empirically from the class label frequency in the training data or from prior knowledge.

Our formulation has been motivated and modified from the original SLA (Tai et al., 2021) to introduce the lower bound $\mathbf{w}_-$ specifying the minimum number of data points to be assigned in each class. We refer to Appendix B.6 for the ablation study on the effect of this lower bound and Appendix A.2 for the derivations on the equivalence between the inequality in Eqs. (6,7,8,9) for linear programming and equality in Eqs. (11,12,13,14,15) for optimal transport.

We define the marginal distributions for row $\mathbf{r} = [\mathbf{1}_N^T; N(\mathbf{w}_+^T \mathbf{1}_k - \rho \mathbf{w}_-^T \mathbf{1}_K)]^T \in \mathbb{R}^{N+1}$ and column $\mathbf{c} = [\mathbf{w}_+ N; N(1 - \rho \mathbf{w}_-^T \mathbf{1}_K)]^T \in \mathbb{R}^{K+1}$. We define the prediction matrix over the unlabeled set, which satisfied the confidence test, $P \in \mathbb{R}^{M \times N \times K}$ such that each element $P_{m,i,k} = p_m(y = k \mid \mathbf{x}_i)$ and the averaging prediction over $M$ models is $\bar{P} = \frac{1}{M} \sum_{m=1}^{M} P_{m,*,*} \in \mathbb{R}^{N \times K}$. The cost and the assignment matrices are defined below

$$C := \begin{bmatrix} -\log \bar{P} & \mathbf{0}_N \\ \mathbf{0}_K^T & 0 \end{bmatrix} \in \mathbb{R}^{(N+1) \times (K+1)} \quad (16) \qquad \tilde{Q} := \begin{bmatrix} Q & \mathbf{u} \\ \mathbf{v}^T & \tau \end{bmatrix} \in \mathbb{R}^{(N+1) \times (K+1)}. \quad (17)$$

We derive in Appendix A.3 the optimization process to learn $Q$ by initializing $b^{(j)} = \mathbf{1}_K^T$ and iteratively updating for $j = 1...J$ iterations or until convergence:

$$a^{(j+1)} = \frac{\mathbf{r}}{\exp\left(-\frac{C_{i,k}}{\varepsilon}\right) b^{(j)}} \quad (18) \qquad b^{(j+1)} = \frac{\mathbf{c}}{\exp\left(-\frac{C_{i,k}}{\varepsilon}\right)^T a^{(j+1)}}. \quad (19)$$

After estimating $\tilde{Q}$, we get $Q$ from Eq. (17) and assign labels to unlabeled data, i.e., where $Q_{i,k} > 0$, and repeat the whole pseudo-labeling process for a few iterations as shown in Algorithm 1. The optimization process above can also be estimated using mini-batches (Fatras et al., 2020) if needed.

**Discussion.** Instead of performing greedy selection using a threshold $\gamma$ like other PL algorithms, our CSA specifies the frequency of assigned labels including the lower bound $\mathbf{w}_-$ and upper bound $\mathbf{w}_+$ per class as well as the fraction of data points $\rho \in (0, 1)$ to be assigned. Then, the optimal transport will automatically perform row and column scalings to find the 'optimal' assignments such that the selected element, $Q_{i,k} > 0$, is among the highest values in the row (within a data point $i$-th) and at the

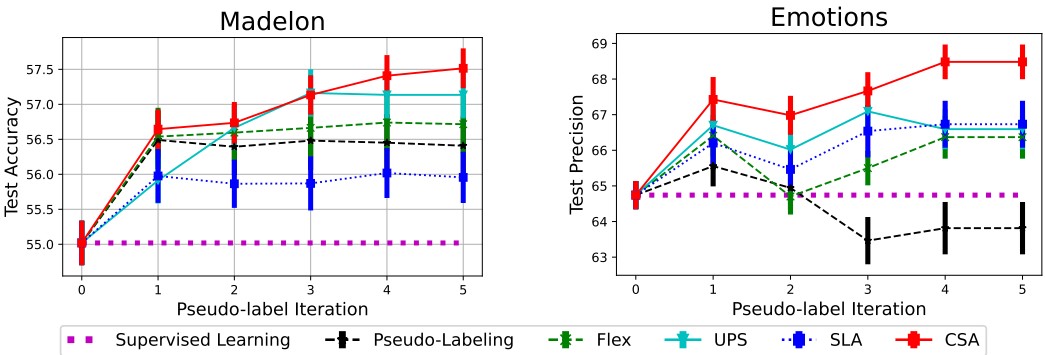

Figure 3: Comparison with the pseudo-labeling methods on tabular data for single-label classification (*left*) and multi-label classification (*right*). x-axis indicates the iteration $t = 1, ..., T$.

same time receive the highest values in the column (within a class $k$-th). In contrast, existing PL will either look at the highest values in row or column, separately – but not jointly like ours. Here, the high value refers to high probability $P$ or low-cost $C$ in Eq. (5).

**Multi-label.** The resulting matrix $Q$ can naturally assign each data point to multiple classes. By that way, we can perform multi-label classification (Read et al., 2011; Nguyen et al., 2016) by assigning a vector of labels $k$ to each data point $i$ s.t. $Q_{i,k} > 0$. This multi-label option is also readily available for other pseudo-labeling methods (Lee et al., 2013). In this multi-label setting, we will use the total variance as the main criteria for confidence estimation as the T-test is not suitable to consider the co-existence of multiple labels.

# 3 EXPERIMENTS

**Baselines.** We select to compare our CSA with the following baselines which are suitable for the data setting without data augmentation: Pseudo-labeling (PL) (Lee et al., 2013), FlexMatch (Zhang et al., 2021), UPS (Rizve et al., 2021) and SLA (Tai et al., 2021). We adapt the label assignment mechanisms in Zhang et al. (2021); Rizve et al. (2021) for tabular domains although their original designs are for computer vision tasks. Without explicitly stated, we use a pre-defined threshold of 0.8 for PL, FlexMatch, UPS. We further vary this threshold with different values in Section 3.2. We also compare with the supervised learning method – trained using the labeled data. The multi-layer perceptron (MLP) implemented in Yoon et al. (2020) includes two layers using 100 hidden units. Additional to these pseudo-labeling based approaches, we compare the proposed method with Vime.

**Data.** We use public datasets for single-label and multi-label classifications from UCI repository (Asuncion & Newman, 2007). Since we do not use any domain assumption, these data are presented as vectorized or tabular data formats. In addition, we obtain three products, renamed as A, B, and C from Amazon catalog. The Amazon data include product numerical and categorical attributes as well as text features of a product review. We summarize all dataset statistics in Appendix Table 7. Our data covers various domains including: image, language, medical, biology, audio, finance and retail.

**Classifier and hyperparameters.** Given limited labeled data presented in tabular format, we choose XGBoost (Chen & Guestrin, 2016) as the main classifier which typically outperforms state-of-the-art deep learning approaches in this setting (Shwartz-Ziv & Armon, 2022) although our method is more general and not restricted to XGBoost. We use $M = 20$ XGBoost models for ensembling. We refer to Appendix B.3 on the empirical analysis with different choices of $M$. The ranges for these hyperparameters are summarized in Appendix Table 6.

**Setting.** Given the setting with limited labels data, we do not use a validation set for tuning hyperparameters of the XGBoost model. We repeat the experiments 30 times with different random seeds, then report the mean and standard error. All Python implementations will be released in the final version.

## 3.1 COMPARISON OF PSEUDO-LABELING FOR SINGLE AND MULTI-LABEL CLASSIFICATION

We use accuracy as the main metric for evaluating single-label and precision for multi-label classification (Wu & Zhou, 2017). We first compare our CSA with the methods in pseudo-labeling family.

Table 2: Comparison with pseudo-labeling methods.

| Datasets | Supervised Learning | | Vime | PL | Flex | UPS | SLA | CSA |
|---|---|---|---|---|---|---|---|---|
| | XGBoost | MLP | | | | | | |
| Segment | 95.42±1 | 94.63±1 | 92.71±1 | 95.68±1 | 95.68±1 | 95.67±1 | 95.80±1 | **95.90**±1 |
| Wdbc | 89.20±3 | 91.33±2 | **91.83**±5 | 91.23±3 | 91.23±3 | 91.62±3 | 90.61±2 | **91.83**±3 |
| Analcatdata | 91.63±2 | 96.17±1 | 96.13±2 | 90.95±2 | 90.62±3 | 91.33±3 | 90.98±2 | **96.60**±2 |
| German-credit | 70.73±2 | 71.10±3 | 70.50±4 | 70.72±3 | 70.72±3 | 71.15±2 | 70.72±3 | **71.47**±3 |
| Madelon | 54.80±3 | 50.80±2 | 52.97±2 | 56.45±4 | 56.74±4 | 57.13±3 | 56.53±4 | **57.51**±3 |
| Dna | 88.53±1 | 76.50±2 | 79.07±3 | 88.17±1 | 88.17±1 | 88.51±1 | 88.09±2 | **89.24**±1 |
| Agar Lepiota | 57.63±1 | 63.80±1 | **63.83**±1 | 58.98±1 | 59.53±1 | 58.88±1 | 58.96±1 | 59.53±1 |
| Breast cancer | 93.20±2 | 86.40±6 | 85.87±5 | 92.89±2 | 92.89±2 | 93.38±2 | 92.76±2 | **93.55**±2 |
| Digits | 82.23±3 | 86.80±1 | 84.10±1 | 81.67±3 | 81.44±3 | 83.78±3 | 81.51±3 | **88.10**±2 |
| Category A | 72.71±3 | 57.15±9 | 50.55±6 | 72.48±3 | 72.35±3 | 72.71±3 | 72.24±3 | **73.97**±4 |
| Category B | 79.48±3 | 42.85±2 | 36.45±4 | 79.41±2 | 80.49±1 | 79.45±2 | 79.82±2 | **80.62**±2 |
| Category C | 61.35±1 | 51.90±1 | 52.80±1 | 61.23±1 | 61.98±1 | 61.65±1 | 62.04±1 | **63.01**±1 |

We show the performance achieved on the held-out test set as we iteratively assign pseudo-label to the unlabeled samples, i.e., varying $t = 1, ..., T$. As presented in Fig. 3 and Appendix Fig. 11, CSA significantly outperforms the baselines by a wide margin in the datasets of Analcatdata, Synthetic Control and Digits. CSA improves approximately 6% compared with fully supervised learning. CSA gains 2% in Madelon dataset and in the range of [0.4%, 0.8%] on other datasets.

Unlike other pseudo-labeling methods (PL, FlexMatch, UPS), CSA can get rid of the requirement to predefine the suitable threshold $\gamma \in [0, 1]$, which is unknown in advance and should vary with different datasets. In addition, the selection in CSA is non-greedy that considers the relative importance within and across rows and columns by the Sinkhorn's algorithm while the existing PL methods can be greedy when only compared against a predefined threshold.

Furthermore, our CSA outperforms SLA (Tai et al., 2021), which does not take into consideration the confidence score. Thus, SLA may assign labels to 'uncertain' samples which can degrade the performance. From the comparison with SLA, we have shown that uncertainty estimation provides us a tool to identify susceptible points to ignore from the data and thus can be beneficial for SSL.

We then compare our CSA with Vime (Yoon et al., 2020), the current state-of-the-art methods in semi-supervised learning for tabular data in Table 2. CSA performs better than Vime in most cases, except the Agaricus Lepiota dataset which has an adequately large size (6500 samples, see Table 7) – thus the deep network component in Vime can successfully learn and perform well.

**Multi-label classification.** The advantage of CSA is further demonstrated to assign labels in multi-label classification settings. We present the comparison in the bottom row of Fig. 3 where we show that CSA performs competitively and better than the baselines on the three datasets considered.

## 3.2    COMPARISON OF CSA AGAINST PSEUDO-LABELING WITH DIFFERENT THRESHOLDS

One of the key advantages of using optimal transport is that it can learn the non-greedy assignment by performing row-scaling and column-scaling to get the best assignment such that the cost $C$ is minimized, or the likelihood is maximized with respect to the given constraints from the row **r** and column **c** marginal distributions. By doing this, our CSA can get rid of the necessity to define the threshold $\gamma$ and perform greedy selection in most of the existing pseudo-labeling methods.

We may be wondering if the optimal transport assignment $Q$ can be achieved by simply varying the thresholds $\gamma$ in pseudo-labeling methods? The answer is no. To back up our claim, we perform two analyses. First, we visualize and compare the assignment matrix achieved by CSA and by varying a threshold $\gamma \in \{0.5, 0.7, 0.9\}$ in PL. As shown in *left* Fig. 4, the outputs are different and there are some assignments which could not be selected by PL with varying $\gamma$, *eg.*, the following {row index, column index}: {1, 3}, {9, 3} and {14, 3} annotated in red square in *left* Fig. 4.

Second, we empirically compare the performance of CSA against varying $\gamma \in \{0.7, 0.8, 0.9, 0.95\}$ in PL. We present the results in *right* Fig. 4, which again validates our claim that the optimal assignment in CSA will lead to better performance consistently than PL with changing values of $\gamma$.

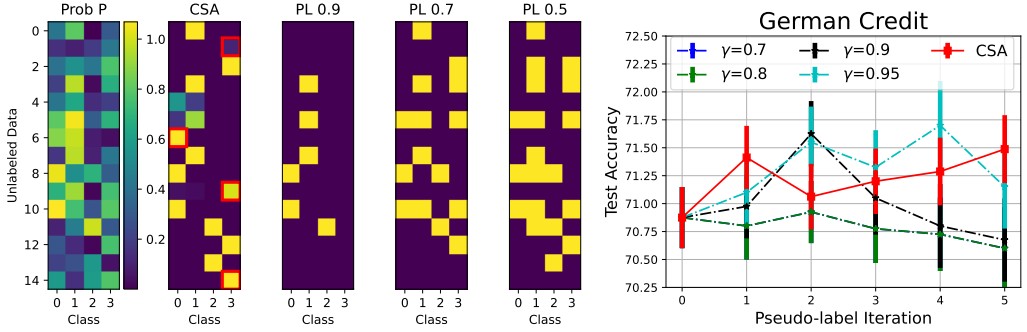

Figure 4: Comparison between CSA versus pseudo-labeling with different thresholds $\gamma$. *Left*: the assignments by CSA can not be simply achieved by varying the threshold in PL, *eg.* see the locations highlighted in red square. *Right*: Comparison when varying thresholds in PL against CSA.

### 3.3 ABLATION STUDIES

**Varying the number of labels and unlabels.** Different approaches exhibit substantially different levels of sensitivity to the amount of labeled and unlabeled data, as mentioned in Oliver et al. (2018). We, therefore, validate our claim in Theorem 1 by empirically demonstrating the model performance with increasing the number of unlabeled examples. In *left* Fig. 5, we show that the model will be beneficial with increasing the number of unlabeled data points.

**Limitation.** In some situations, such as when labeled data is too small in size or contains outliers, our CSA will likely assign incorrect labels to the unlabeled points at the earlier iteration. This erroneous will be accumulated further and lead to poor performance, as shown in *left* Fig. 5 when the number of labels is 100 and the number of unlabels is less than 500. We note, however, that other pseudo-labeling methods will share the same limitation.

**Further analysis.** We refer to Appendix B for other empirical analysis which can be sensitive to the performance, including (i) different choices of pseudo-labeling iterations $T$ and (ii) different number of choices for XGB models $M$, (iii) computational time for each component and (iv) statistics on the number of points selected by each component per iteration.

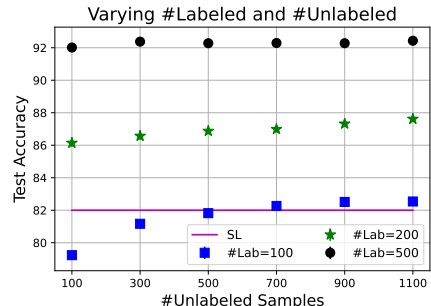

Figure 5: Performance on Digit w.r.t. the number of unlabeled samples given the number of labeled samples as $100, 200, 500$, respectively.

## 4 CONCLUSION AND FUTURE WORK

Although there has been significant progress in SSL for images and text domains, limited attention has been invested for unstructured domains in which data augmentation or pretraining is not available, such as data presenting in tabular format. We propose CSA, a new method for pseudo-labeling particularly suitable for tabular domains but not restricted to.

Our CSA has two key ingredients. First, it estimates and takes into account the confidence levels in assigning labels. Second, it learns the optimal assignment using Sinkhorn algorithm which appropriately scales rows and columns probability to get assignment without the greedy selection using a threshold as used in most of PL methods.

The proposed CSA maintains the benefits of PL in its simplicity, generality, and ease of implementation while CSA can significantly improve PL performance by addressing the overconfidence issue and better assignment with optimal transport.

Future work can extend the ensembling process by not only using XGBoost, but also other classifiers, such as AdaBoost (Wyner et al., 2017) or CatBoost (Prokhorenkova et al., 2018). Another extension is to use CSA as the main label assignment method for SSL and integrate it into training deep learning model to build CSAMatch, analogous to FixMatch (Sohn et al., 2020), MixMatch (Berthelot et al., 2019) and FlexMatch (Zhang et al., 2021).

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
