# OpenReview forum: "Confident Sinkhorn Allocation for Pseudo-Labeling"
_ICLR.cc/2023/Conference — Submitted to ICLR 2023_

### Official Review · Reviewer_hVwg · 2022-10-15

**Confidence:** 4
**Correctness:** 3
**Technical Novelty And Significance:** 3
**Empirical Novelty And Significance:** 2
**Recommendation:** 5

**Clarity, Quality, Novelty And Reproducibility:**

As mentioned previously, I think the paper is generally well written. The details of the proposed algorithm are clearly stated.

**Strength And Weaknesses:**

(+) The paper is generally well-written.

(+) The theoretical analyses of this paper are solid.

(+) The experiments are sufficient.


(-) I feel that the proposed method is computational expensive. In each iteration, M models should be trained on labeled data, and an inner loop exists for Sinkhorn’s algorithm. Therefore, I want to see the total running time comparison of the proposed CSA and baseline methods. Besides, how about the scalability of the proposed method?

(-) Also regarding the model assembling, I doubt that the performance improvement of CSA over other comparators in Table 2 is due to this operation. Moreover, I found that the performance improvement on various dataset is quite marginal. Therefore, I want to know whether the comparison is fair, and whether the comparison demonstrates the effectiveness of the proposed way for calculating the uncertainty. Maybe, the authors should conduct statistical significance test to justify the superiority.


**Summary Of The Paper:**

This paper presents a novel problem for pseudo-labeling based semi-supervised learning. Especially, the authors proposed Confident Sinkhorn Allocation (CSA), which assigns labels to only examples with high confidence scores and learns the best label allocation via optimal transport.

**Summary Of The Review:**

I like the solid theoretical analyses of this paper, although the experiments are comparatively weak. I would temporarily give a positive mark for this paper.

---

> ### Author Response · Authors · 2022-11-10
> **Author's response**
>
> Thank you for the overall positive review. We are glad you found the work to be well-written, solid theoretical analysis and sufficient experimental results. Please see our response below.
>
> ---
> **Q1**:
> In each iteration, M models should be trained on labeled data, and an inner loop exists for Sinkhorn’s algorithm. Therefore, I want to see the total running time comparison of the proposed CSA and baseline methods.
>
> *Answer*:
> The computational cost for CSA is presented in Appendix B4. We can see that the computational cost for optimal transport is negligible as shown in Fig 9 in the Appendix. It takes approximately 0.1 second for Madelon dataset (K=2, d=500, #labeled=124, #unlabelled=1956).
>
> Although the ensemble approach for uncertainty estimation will require more computation, it is still not very expensive, such as in the Madelon dataset, the XGB model training takes 2 seconds for each XGB model. This means the total time is 40 seconds for training M=20 models. It is important to note that this computation depends on the size of the labeled data which is typically small in most pseudo-labelling applications.
>
> ---
>
> **Q2**: I found that the performance improvement on various datasets is quite marginal. Therefore, I want to know whether the comparison is fair, and whether the comparison demonstrates the effectiveness of the proposed way for calculating the uncertainty. Maybe, the authors should conduct statistical significance test to justify the superiority.
>
> *Answer*: The effectiveness of the uncertainty can be seen from the Appendix B1 and Table 4.
> Our pseudo-labeling algorithm will try to improve the performance on top of the existing supervised learning algorithms (XGBoost and MLP) which are somewhat already effective. Therefore, we may not expect a significant boost in the performance.
> Having said that, we can see, for example: the improvement of Analcatdata is 5%, Digits is 6%, Madelon, Wdbc are 2-3% and the remaining datasets are ~1%.
> We thank the reviewer for the suggestion on the statistical significance test. We will consider it in the final version.

---

### Official Review · Reviewer_MB4o · 2022-10-22

**Confidence:** 4
**Correctness:** 4
**Technical Novelty And Significance:** 2
**Empirical Novelty And Significance:** 2
**Recommendation:** 5

**Clarity, Quality, Novelty And Reproducibility:**

The paper is clearly written.

The novelty is limited.

The results seem to be reproducible.

**Strength And Weaknesses:**

**Strengths**:

* The paper proposes a method that does not need a predefined threshold to filter the unlabeled data


**Weakness**:

* The proposed method has limited novelty. First, considering uncertainty in pseudo-labeling is not new, e.g. [2] considered this scheme. Second, using optimal transport is not new as well. It is proposed in SLA [1].


* The proposed method does not need a predefined threshold to filter the unlabeled data, but the method needs additional hyper-parameters, such as fraction of assigned label $\rho$, which counteract the benefit of the proposed methods.


* The proposed methods need additional computation cost due to optimal transport calculation and multiple model training on the labeled data.


References:

[1] Sinkhorn label allocation: Semi-supervised classification via annealed self-training. ICML.2021

[2] In defense of pseudo labeling: An uncertainty-aware pseudo-label selection framework for semi-supervised learning. ICLR 2021

**Summary Of The Paper:**

This paper proposes an improved pseudo-labeling method for semi-supervised learning methods. The paper first proposes a method for filtering out some unlabeled data that has high uncertainty based on  Welch’s T-test and  Total variance.  The paper then extends the methods of SLA [1] to assign labels for unlabeled data.




**Summary Of The Review:**

This paper proposes an improved pseudo-labeling method for semi-supervised learning methods, but the novelty of the proposed methods is limited.

---

> ### Author Response · Authors · 2022-11-10
> **Author's response**
>
> Thank you for the detailed review. We hope we can clarify your issues and provide you with sufficient confidence to raise your score. Please see our responses below.
>
> ---
>
> **Q1**: The proposed method does not need a predefined threshold to filter the unlabeled data, but the method needs additional hyper-parameters, such as fraction of assigned label ρ which counteract the benefit of the proposed methods.
>
> *Answer*: It is beneficial to replace the predefined threshold in pseudo-labeling by the fraction of assigned label as in our paper because:
>
> Using predefined threshold in pseudo-labeling:
> Specifying multiple thresholds, each threshold for a single class, is difficult in practice.
> Specifying a single threshold for all classes, this will make the assignment biased toward the dominant class (e.g., when the classes have different proportions).
>
> On the other hand, if we specify the fraction of assigned labels as in our method, the optimal transport will learn the best assignment to satisfy this constraint: without suffering the imbalance bias and without the need to specify multiple thresholds.
>
> ---
>
> **Q2**: The proposed methods need additional computation cost due to optimal transport calculation and multiple model training on the labeled data.
>
> *Answer:* The computational cost for CSA is presented in Appendix B4. We can see that the computational cost for optimal transport is negligible as shown in Appendix Fig 9. It takes approximately 0.1 second for Madelon dataset (K=2, d=500, #labeled=124, #unlabelled=1956).
>
> Although the ensemble approach for uncertainty estimation will require more computation, it is still not very expensive, such as in the Madelon dataset, the XGB model training takes 2 seconds for each XGB model. This means the total time is 40 seconds for training M=20 models. It is important to note that this computation depends on the size of the labeled data which is typically small in most pseudo-labelling applications.
>
> ---
> **Q3**: The proposed method has limited novelty. First, considering uncertainty in pseudo-labeling is not new, e.g. [2] considered this scheme. Second, using optimal transport is not new as well. It is proposed in SLA [1].
>
> *Answer*: Our first contribution is Theorem 1 to study the important role of uncertainty in obtaining a good classifier for a semi-supervised learning approach. It is novel because we are the first to study this property of the uncertainty in the community, to the best of our knowledge.
>
> We have proposed to use T-test for uncertainty estimation to discard the low confidence data points and by doing that it gets rid of the need to specify the predefined threshold for pseudo-labeling. This T-test for pseudo-labeling is a novel contribution. See the Appendix Table 4 for the empirical gain using T-test versus without T-test in the original SLA [1].
>
> We admit that we are not the first to use optimal transport for pseudo-labeling. However, we did make optimal transport better with introducing the lower bound w_. Again, see Appendix Table 4 for the improvement with the lower bound w_ versus the original SLA [1].
>
> ---
> We feel we have addressed all of the concerns raised. If this is not the case, please let us know so that we can have the opportunity to discuss further.

---

### Official Review · Reviewer_AnF4 · 2022-10-25

**Confidence:** 4
**Correctness:** 3
**Technical Novelty And Significance:** 3
**Empirical Novelty And Significance:** 2
**Recommendation:** 5

**Clarity, Quality, Novelty And Reproducibility:**

This paper is overall good and well presented. I personally would not encourage more research efforts on improving pseudo-labeling. The major reason is that studying this specific topic for SSL is narrow because according to the experimental results, such an elaborate design only slightly enhances the SSL performance. It is okay for the method such as FixMatch back to two years ago, but it is not very interesting today.

**Strength And Weaknesses:**

Strength:
1. CSA first conducts sample selection by removing low-confidence samples and then focuses on assigning labels. The overall framework is convincing and easy to follow.
2. The variance of correct assignment, i.e., epistemic uncertainty, is considered for evaluating the uncertainty, which is an interesting idea.

Weaknesses:
1. The so-called domain-specific and pseudo-labeling has been attributed to two separate components which were thought based on the different designing principles. Please refer to [1].
2. I understand that authors would like to show correct assignments affect the optimal decision boundary by theorem 1. As it extends Yang&Xu’s theory, I am curious if it gives a better theoretical conclusion, e.g., closer to the decision boundary.
3. The discussion about the most provable class and second-most probable class is related to the theorem presented in the work [1], because both considered the relative relation between outputs.
4. T-test for sample selection is based on training M models, which incurs much computation cost. Note that the dropout used in Rizves et al is only involving inference, but CSA is with an ensemble style. An explanation about it should be added.
5. Another concern is that T-test is used for sample selection, but it has not used epistemic uncertainties, i.e., the variance of I. Notice that total variance is designed for multilabel classification, is it feasible for multiclass semi-supervised classification?
6. The contribution compared with Tai et al from the label assignment section is not very convincing to me. Does CSA outperform theirs by T-test and some additional constraints over class population size?

[1] Taming Overconfident Prediction on Unlabeled Data from Hindsight, https://arxiv.org/abs/2112.08200



**Summary Of The Paper:**

This paper aims to improve the well-known pseudo-labeling by Confident Sinkhorn Allocation (CSA) which confidently assigns the labels via optimal transport. The authors demonstrated CSA’s performance by comparing it with other SSL strategies.

**Summary Of The Review:**

I have listed my concerns about this work above and I am expecting the authors' response.

---

> ### Author Response · Authors · 2022-11-10
> **Author's response**
>
> We thank the reviewer for insightful suggestions which are very helpful to improve the clarity of the paper. There are encouraging comments which we appreciate.
>
> ---
>
> **Q1**: The so-called domain-specific and pseudo-labeling has been attributed to two separate components which were thought based on the different designing principles in [1]. The discussion about the most provable class and second-most probable class is related to the theorem presented in the work [1].
> [1] Taming Overconfident Prediction on Unlabeled Data from Hindsight, https://arxiv.org/abs/2112.08200
>
> *Answer*: We thank the reviewer for the suggestion. We will acknowledge and mention [1] which is the related work to our paper.
>
> ---
>
> **Q2**: T-test for sample selection is based on training M models, which incurs much computation cost. Note that the dropout used in Rizves et al is only involving inference, but CSA is with an ensemble style. An explanation about it should be added.
>
> *Answer*: The previous works in semi-supervised learning (such as Rizves et al 2021) focus on deep learning applications in which they can utilize dropout for uncertainty estimation.
>
> On the other hand, our paper focuses on the tabular domains where XGB and ensemble learning algorithms are more suitable. Therefore, we are unable to use dropout (since it is for deep learning approaches), instead we have used an ensemble technique which involves additional computational cost, as presented in the appendix B.4. However, we would highlight that the extra computation for ensemble learning is because of the setting considered for tabular domains.
>
> ---
>
> **Q3**: The contribution compared with Tai et al from the label assignment section is not very convincing to me. Does CSA outperform theirs by T-test and some additional constraints over class population size?
>
> *Answer*: We study the effect of introducing the lower bound w- for the optimal transport and how it improves upon Tai et al formulation in the Appendix B6 and Table 5. We analyze the contribution of each component separately: the lower bound constraint and uncertainty estimation toward the final performance of CSA. Through this analysis, we show that CSA outperforms Tai et al in either ways: (i) using lower bound w- without the T-test, or (ii) using lower bound w- and with the T-test.
>
> ---
>
> **Q4**: Notice that total variance is designed for multilabel classification, is it feasible for multiclass semi-supervised classification?
>
> *Answer*: the total variance can also be used for multiclass semi-supervised classification.  We have presented this setting in the appendix B1 and Table 4. We show that while total variance can be used to multiclass settings, the performance of total variance is not as good as using T-test.
>
> ---
>
> **Q5**: I understand that authors would like to show correct assignments affect the optimal decision boundary by theorem 1. As it extends Yang&Xu’s theory, I am curious if it gives a better theoretical conclusion, e.g., closer to the decision boundary.
>
> *Answer*:  The main goal of our theory is to highlight the important role of uncertainty in obtaining the good classifier which has not been considered in Yang&Xu’s analysis.
>
> ---
> **Q6**: Another concern is that T-test is used for sample selection, but it has not used epistemic uncertainties, i.e., the variance of I.
>
> *Answer*: the epistemic uncertainty indicates how the model is unsure in the prediction. This epistemic uncertainty is encoded in the two empirical distributions for T-test, representing how multiple XGB models are uncertain in making the prediction.
>
> ---
>
> We believe that our responses above have addressed your concern to increase a score. If not, we again appreciate the opportunity to clarify additional areas of concern.

---

> > ### Comment · Reviewer_AnF4 · 2022-11-29
> > **Comments after reading the response**
> >
> > First of all, I would appreciate the authors' response. The part where you mentioned that this paper focuses on tabular data sort of addressed my concerns. However, It also makes me lower my confidence score, as you considered some special data that hinders existing SSL techniques. Regarding the computation issue which was also mentioned by other reviewers, I felt it might be case dependent although you have shown the results on some small datasets in Appendix B.4. My biggest concern right now is still the limitation of novelty. After reading again your summarized contribution, I am afraid that I could not give you a very positive comment based on the current version.

---

### Decision · Program_Chairs · 2023-01-20

**Decision:**

Reject

**Justification For Why Not Higher Score:**

The main issue is the limited novelty given the existence of related methods.

**Justification For Why Not Lower Score:**

N/A

**Metareview: Summary, Strengths And Weaknesses:**

The paper studied pseudo-labeling for semi-supervised learning and proposed a better uncertainty-aware method. The main issue is the limited novelty given the existence of related methods, for example, domain-specific and pseudo-labeling has been attributed to two separate components which were thought based on the different designing principles. Moreover, the significance may also be an issue: the paper focused on tabular data, the computation should be expensive, and the improvement looked marginal. The rebuttals by the authors didn't successfully resolve the concerns from the reviewers, especially the novelty issue. Consequently, we cannot accept it for publication. If the authors would like to revise the paper and try another machine learning conference, I suggest the authors to improve the quality of experiments in the revised manuscript (even better if the novelty can be further improved).